# The Isolation and Structure Elucidation of Spirotetronate Lobophorins A, B, and H8 from *Streptomyces* sp. CB09030 and Their Biosynthetic Gene Cluster

**DOI:** 10.3390/molecules28083597

**Published:** 2023-04-20

**Authors:** Jie Shi, Dian Peng, Fei-Fei Peng, Qing-Bo Zhang, Yan-Wen Duan, Yong Huang

**Affiliations:** 1Xiangya International Academy of Translational Medicine, Central South University, Changsha 410013, China; 2School of Pharmaceutical Sciences, Changsha Health Vocational College, Changsha 410100, China; 3South China Sea Institute of Oceanology, Chinese Academy of Sciences, Guangzhou 510301, China; 4Hunan Engineering Research Center of Combinatorial Biosynthesis and Natural Product Drug Discovery, Changsha 410011, China; 5National Engineering Research Center of Combinatorial Biosynthesis for Drug Discovery, Changsha 410011, China; 6Institute of Health and Medicine, Hefei Comprehensive National Science Center, Hefei 230093, China

**Keywords:** *Streptomyces*, spirotetronate, lobophorins, transwell, anti-mycobacterial activity

## Abstract

Lobophorins (LOBs) are a growing family of spirotetronate natural products with significant cytotoxicity, anti-inflammatory, and antibacterial activities. Herein, we report the transwell-based discovery of *Streptomyces* sp. CB09030 from a panel of 16 in-house *Streptomyces* strains, which has significant anti-mycobacterial activity and produces LOB A (**1**), LOB B (**2**), and LOB H8 (**3**). Genome sequencing and bioinformatic analyses revealed the potential biosynthetic gene cluster (BGC) for **1**–**3**, which is highly homologous with the reported BGCs for LOBs. However, the glycosyltransferase LobG1 in *S*. sp. CB09030 has certain point mutations compared to the reported LobG1. Finally, LOB analogue **4** (O-β-D-kijanosyl-(1→17)-kijanolide) was obtained through an acid-catalyzed hydrolysis of **2**. Compounds **1**–**4** showed different antibacterial activities against *Mycobacterium smegmatis* and *Bacillus subtilis*, which revealed the varying roles of different sugars in their antibacterial activities.

## 1. Introduction

Spirotetronate polyketides have displayed a variety of impressive biological activities, such as a significant cytotoxicity against tested tumor cell lines (spirohexenolide A (**a**) [1], okilactomycin (**b**) [2], and chrolactomycin (**c**) [3,4]), as well as anti-inflammatory and antibacterial activities (abyssomicin C (**d**), kijanimicin (**e**), lobophorin B (**f**), and MM46115 (**g**)) (Figure 1) [5,6,7,8]. Structurally, the over 70-membered spirotetronates are classified into class I, which features the spirotetronate moiety within a macrocycle with different sizes, or class II with an additional decalin. Lobophorins (LOBs) belong to class II spirotetronates and share several structural characteristics with kijanimicin, such as a decalin-containing pentacyclic aglycon, a D-kijanose or D-kijanose-like sugar, and an oligosaccharide moiety. LOB A and B, the first two members of LOBs, were reported by Fenical and co-workers from *Actinomyces* sp. CNB-837, a marine bacterium living in symbiosis with the Caribbean brown alga *Lobophora variegate* in 1999 [8]. To our knowledge, 16 LOBs have been isolated, including LOBs A–L (**1**, **2**, **30**, **31**, and **5**–**12**) [8,9,10,11,12,13,14,15,16] and LOBs CR1-CR4 (**13**, **14**, **32**, and **33**) (Figure 1) [17,18]. Furthermore, 16 LOB derivatives, including LOBs H1-H15 (**3** and **15**–**28**) and LOB G2-1 (**29**) (Figure 1), were subsequently produced by the heterologous expression of the LOB biosynthetic gene cluster (BGC) from the marine-derived *Streptomyces pactum* SCSIO 02999 by Zhang and colleagues [19]. Recently, three derivative LOBs, N1–N3 (**34**–**36**), were isolated from the mutant by the disruption of the P450 monooxygenase LobP1 gene from *Streptomyces* sp. SCSIO 01127 [20].

Transwell-based microbial culturing systems have recently been used to co-culture and discover new natural products from several bacterial strains. In transwell, bacteria grow on the upper agar layer, which draws nutrients from the liquid media in the bottom layer and also allows for the diffusion of metabolites from the agar. For example, Clardy [21] and co-workers co-cultured nine actinomycetes in transwell plates, which allowed for the continuous monitoring of the metabolite production without disturbing the co-culturing community and led to the isolation of a new antibiotic, amycomicin [22]. Similarly, conocandins and dentigerumycin were discovered [23].

There is a great interest in discovering natural products to use against *Mycobacterium ulcerans*, the causative agent of Buruli ulcer [24,25]. This is the third most common mycobacterium disease after tuberculosis and leprosy [26]. *M. ulcerans* causes a serious skin infection and then induces a serious inflammation by secreting a macrolide toxin called mycolactone [24,27]. In this study, we hypothesize that the transwell assay system may allow for the screening of our in-house actinomycete strain collection and discover the strains with promising anti-mycobacterial activity. *Streptomyces* sp. CB09030 was thus discovered from a panel of 16 in-house *Streptomyces* strains, which produces LOB A (**1**), LOB B (**2**), and LOB H8 (**3**). A bioinformatic analysis of the LOB BGC revealed a glycosyltransferase LobG1, with several point mutations that were different from the previously reported glycosyltransferase responsible for D-kijanose attachment. The LOB analogue **4** (O-β-d-kijanosyl-(1→17)-kijanolide) was also obtained through an acid-catalyzed hydrolysis of **2**. The study of **1**–**4** revealed the essential role of the oligosaccharide moiety in the antibacterial activities of **1** and **2** against *Mycobacterium smegmatis* MC^2^ 155, the fast-growing model strain of mycobacterium, and *Bacillus subtilis* 62305.

## 2. Results and Discussion

### 2.1. A Transwell-Based Culture Bioassay to Discover Streptomyces *sp.* CB09030 with Anti-Mycobacterial Potential

Due to the simplicity of performing a bioassay on the secreted secondary metabolites from *Streptomyces* strains growing on the solid agar, we cultured 16 *Streptomyces* species using the liquid medium from the transwell, with *Mycobacterium smegmatis* MC^2^ 155 as the indicating strain for anti-mycobacterial activity. Among them, *S.* sp. CB0657, *S*. sp. CB01580, *S*. sp. CB02366, and *S*. sp. CB09030 showed certain inhibitory activity against *M. smegmatis* MC^2^ 155 (Appendix A). Further HPLC analyses of the culture extracts of these four strains in 250 mL shaking flasks revealed that *S*. sp. CB09030 produced more secondary metabolites than the other strains (Appendix A). Therefore, it was used for a scaled-up fermentation to isolate the potential natural products with significant anti-mycobacterial activities.

### 2.2. Isolation and Structure Elucidation of ***1***–***3*** from S. *sp.* CB09030

A large scale fermentation of *S*. sp. CB09030 (27 L) led to the production and isolation of compounds **1**–**3** (Figure 2A). Compound **1** was isolated as a white powder. HR-ESI-MS revealed its molecular formula to be C_61_H_92_N_2_O_19_, based on the molecular ions of [M + H]^+^ 1157.6359 (calcd 1157.6370) and [M + Na]^+^ 1179.6178 (calcd 1179.6190) (Appendix A). The NMR data of **1** are almost identical to LOB A, which was previously isolated from *Actinomyces* sp. CNB-837, deep-sea derived *Streptomcyes* sp. SCSIO 01127, and marine-derived *Streptomyces* sp. MS100061 (Appendix A) [8,11,19]. Further HR-ESI-MS/MS analyses revealed the fragment ions belonging to the LOB A aglycon ([M + H − H_2_O]^+^ 517.2952 calcd 517.2954), the aglycon with A and D sugars ([M + H]^+^ 883.4941, calcd 883.4935), and the aglycon with sugar D ([M + H]^+^ 753.4308, calcd 753.4327), which further confirmed that **1** is LOB A.

Compound **2** was isolated as a white powder with a similar UV absorption to **1**, suggesting that **2** might have a similar molecular skeleton to **1**. HR-ESI-MS analyses revealed its molecular formula to be C_61_H_90_N_2_O_21_, based on the molecular ions of [M + Na]^+^ 1209.5921 (calcd 1209.5934) and [M + NH_4_]^+^ 1204.6367 (calcd 1204.6380) (Appendix A). Further HR-ESI-MS/MS analyses revealed the fragment ions belonging to the identical LOB A aglycon ([M + H − H_2_O]^+^ 517.2945 calcd 517.2954) and the different sugar attachments with the fragment ions of [M + H]^+^ 913.4707 (calcd 913.4698) and [M + H]^+^ 783.4075 (calcd 783.4068), respectively. The 1D and 2D NMR spectra of **2** further revealed that **2** is the previously isolated LOB B with an unusual D-kijanose sugar, which was usually isolated with LOB A (Appendix A) [8,11,19].

Compound **3** was also isolated as a white powder. HR-ESI-MS revealed its molecular formula to be C_33_H_44_O_6_, based on the molecular ion of [M − H]^−^ 535.3206 (calcd 535.3060) (Appendix A). Its NMR spectra showed that **3** is the previously isolated LOB H8 via the heterologous expression of the LOB BGC (Appendix A) [19].

### 2.3. Acid Hydrolysis of ***2*** Leads to the Generation of ***4***

In order to obtain more LOB derivatives, **2** was further hydrolyzed in 0.5 M-methanolic HCl to produce **4** (Figure 3A). The molecular formula of **4** was deduced to be C_42_H_58_N_2_O_12_, based on the molecular ion of 783.4076 [M + H]^+^ (calcd 783.4068) (Appendix A). The structure of **4** was established by analyses of its NMR data (Appendix A) to be the previously isolated hydrolysis product of kijanimicin [28]. Based on the comparison of the ^13^C NMR spectra of **2**, **3**, and **4**, only the D-kijanose attached to the aglycon in **4** had a characteristic methine signal at *δ_C_* = 99.0 ppm (Figure 3B). Compound **4** has a hydroxymethyl group attached to its C-22, while a similar LOB derivative, LOB H11, has a methyl group attached to its C-22 from a heterologous host [19].

### 2.4. Bioinformatic Analysis of A Putative LOB BGC from S. sp. CB09030

The co-production of **1**–**3** in *S*. sp. CB09030 is unique among the isolated LOBs-producing *Streptomyces* strains, which usually co-produce **1** and **2**. This suggests that the transfer of the D-kijanose by the putative glycosyltransferase LobG1 to the aglycon **3** is not efficient in this strain under the current culturing conditions. In order to understand the genetic basis, the whole genome sequencing of *S*. sp. CB09030 was used to identify the LOB BGC. The antiSMASH analysis of the *S*. sp. CB09030’s assembled genome sequence revealed that there are dozens of BGCs for the biosynthesis of putative olimycin and xiamycin A, et al. (Appendix A). The putative LOB BGC spans a DNA region of 102 kb, which is highly similar to that of the reported LOB BGC from *S*. sp. SCSIO 01127, with 39 open reading frames (ORFs) (Figure 4A). For example, the five polyketide synthases (PKSs) LobA1–A5 share an ~84% to ~97% sequence identity with those from the LOB BGC in *S*. sp. SCSIO 01127, which was proposed for the assembly of the linear spirotetronate polyketide chain by utilizing six malonyl-CoA, six methylmalonyl-CoA, and a 3-carbon glycerol unit (Figure 4B) [19,29]. A further analysis of the LOB BGC suggested that it is also highly similar to the LOB BGC in *S. olivaceus* SCSIO T05 and the putative LOB BGC in *S*. sp. FXJ 7.023 (Appendix A).

A phylogenetic analysis showed that the LobG1 from *S*. sp. CB09030 is more closely related to the LobG1 in *S.* sp. FXJ 7.023 than those of *S*. sp. SCSIO 01127 and *S. olivaceus* SCSIO T05 (Appendix A). A further amino acid sequence alignment identified several amino acid differences in these glycosyltransferases, among which the LobG1 in *S*. sp. CB09030 had three unique amino acids, S38T, D250G, and V284M (Figure 4C). The importance of these amino acids for the activity of LobG1 in LOB biosynthesis remains to be established, since it is the essential step to decorating the LOB aglycon.

### 2.5. The Antibacterial Activity of ***1***–***4***

Although **1**–**4** have been separately reported in several previous publications, the availability of these compounds in the current study provides an exciting opportunity to study their antibacterial activity. Compound **2** showed a remarkable antibacterial activity against *M. smegmatis* MC^2^ 155, with a minimum inhibitory concentration (MIC) of 4 μg/mL, which is comparable to the clinically used antibiotic rifampicin (Table 1 and Appendix A). Interestingly, compounds **1**, **3**, and **4** only displayed slightly attenuated antibacterial activities against *M. smegmatis* MC^2^ 155, with MICs of 8 μg/mL. In contrast, **2** had an MIC of 1 μg/mL against *B. subtilis* 62305, while **1**, **3**, and **4** only showed weak antibacterial activities against *B. subtilis* 62305, with MICs of 8, 32, and 32 μg/mL, respectively. These obvious differences are consistent with a previous study of **1** and **2** against *M. tuberculosis* H37Rv and *B. subtilis*, which showed MICs of 32 and 16 μg/mL or 12.5 and 1.56 μg/mL, respectively [11,19]. In addition, **1**–**4** showed no antibacterial activities against the tested *Straptococcus aureus* strains.

These data suggest that the oligosaccharide chains in **1** and **2** are critical for the anti-*B. subtilis* activity, while they are almost negligible for their anti-mycobacterial activities. In addition, the presence of D-kijanose has no effect on the antibacterial activities of **3** and **4** against *M. smegmatis* MC^2^ 155 or *B. subtilis*. Since the many sugar moieties present in natural products are responsible for cell recognition, D-kijanose and the oligosaccharide chain in **2** likely play important roles in their interactions with *B. subtilis*, while the anti-mycobacterial activities of **1**–**4** should mainly benefit from the spirotetronate aglycon.

## 3. Materials and Methods

### 3.1. A Transwell-Based Culture Bioassay

In total, sixteen *Streptomyces* strains were used in the current study and were collected and maintained by the National Engineering Research Center of Combinatorial Biosynthesis for Drug Discovery (Changsha, Hunan 410011, PR China). *S*. sp. CB00316, CB00657, CB01249, CB01580, CB01883, CB01950, CB02009rt, CB02056, CB02058, CB02115, CB02130, CB02366, CB02414, CB02460, CB02488rt, and CB09030 were used. Their spores at ~10^6^ CFU/mL were cultured in the upper chamber of transwell plates (400 μL 0.75% agar medium). Gauze’s medium (G1 medium, 2% soluble starch, 0.1% KNO_3_, 0.05% K_2_HPO_4_, 0.05% MgSO_4_·7H_2_O, 0.05% NaCl, and 0.001% FeSO_4_·7H_2_O) (1.5 mL, 10% *v*/*v*) was added to the lower chamber of the transwell for static fermentation. The fermentation broth (10 μL) was taken periodically (3rd to 27th day) from the lower chamber to measure the antibacterial activities against *M. smegmatis* MC^2^ 155 with a paper disk assay.

### 3.2. General Experimental Procedures

The NMR experiments were conducted on a Bruker spectrometer (400 MHz–600 MHz) and the chemical shifts were reported in ppm, relative to CD_3_OD (δ_H_ = 3.31 ppm) for ^1^H NMR and CD_3_OD (δ_C_ = 49.03 ppm) for the ^13^C NMR spectroscopy. Column chromatography (CC) was carried out on silica gel (Yantai Jiangyou Silica Gel Development Co., Ltd., Yantai, China). Semipreparative reversed phase-high-performance liquid chromatography (RP—HPLC) was performed using a Waters 1525 Binary HPLC Pump equipped with a Waters 2489 UV/Visible detector, using a Welch Ultimate AQ—C18 column (250 × 10 mm, 5 μm).

### 3.3. Streptomyces Fermentation in Shaking Flasks and Extraction

The *Streptomyces* spores of *S*. sp. CB0657, *S*. sp. CB001580, *S*. sp. CB02366, and *S*. sp. CB09030 were inoculated into 250 mL shaking flasks that contained 50 mL of TSB liquid medium, which was next cultured for 48 h (30 °C, 200 rpm). The obtained seed cultures were inoculated into 2 L shaking flasks containing 500 mL of G1 medium (containing 3% macroporous resin HP—20) and grown for 7 days (30 °C, 200 rpm). The fermentation cultures were centrifuged (4000 rpm, 10 min) to collect the mycelia and resins.

### 3.4. Isolation and Acid Hydrolysis of ***2***

The mycelia cakes and resin were extracted with MeOH (3 L × 3). A large-scale fermentation (27 L) of the *S*. sp. CB09030 led to the isolation of **1**–**4**. The fatty acid compounds in the crude extract were removed with an n-hexane extraction. The methanol layer was further concentrated under reduced pressure to obtain the crude extract (20 g). The crude extract was subjected to silica gel CC using DCM-EtOAc-MeOH (1:0:0, 9:1:0, 7:3:0, 5:1:0, 3:1:0, 1:1:0, 0:1:0, 0:10:1, 0:9:1, and 0:1:1) to provide eighteen fractions (Fr.1–18). Fr.16 (2.0 g) was separated by silica gel CC eluted with DCM-MeOH (20:1, 15:1, 10:1, 5:1, and 1:1) to yield six subfractions (Fr.16-1–Fr.16-6). Fr.16-1 was further purified by semipreparative RP-HPLC with a flow rate of 3 mL/min and a gradient elution of CH_3_CN/H_2_O (containing 0.1% formic acid) for 25 min (10–95% CH_3_CN for 12 min, followed by 95% for 3 min, and 95–5% for 2 min, followed by 5% for 8 min) to afford compound **1** (9.0 mg). Fr.16-4 was further purified by semipreparative RP-HPLC to afford compounds **2** (22.2 mg) and **3** (3.8 mg). Compound **2** (9.0 mg) was hydrolyzed in 128 µL of 0.5 M-methanolic HCl at 25 °C for 18 h. The reaction system was concentrated by rotary evaporation and then dissolved with an appropriate amount of methanol. Finally, the hydrolytic product **4** (3.4 mg) was separated by semipreparative RP-HPLC.

### 3.5. Genome Sequencing and Analysis

*S*. sp. CB09030 was cultured at 30 °C for 48 h in 50 mL of TSB liquid medium, which contained 0.5% glycine for genomic DNA. A total of 8.4 Mb was generated by Biomarker Technologies (Beijing, China). The biosynthetic gene clusters were predicted using antiSMASH version 6.0.1. In addition, the sequence homology searches method (Blastp) was also used to identify the genes related to biosynthesis. The LOB BGC was confirmed to be in region 22 by a comparison with the BGC in *S*. sp. SCSIO 01127.

### 3.6. Antibacterial Assay

The antibacterial activities of compounds **1**–**4** against *M. smegmatis* MC^2^ 155 and *B. subtilis* 62305 were evaluated using a broth dilution assay [30]. The individual compounds were dissolved and diluted with MeOH to obtain the initial concentrations. The positive control antibiotics, including rifampin and ampicillin, were diluted with DMSO to obtain the initial concentrations. *Bacillus subtilis* 62305 was grown in an LB medium overnight until the OD_600_ reached 0.5–0.6, which were then diluted to 10^6^ CFU/mL. First, the tested compounds or positive controls (100 μL) were mixed with a bacterial solution (100 μL) and stepwise diluted. The plates were incubated at 37 °C overnight. Finally, 50 μL of resazurin was added into each well to visualize the results. Similarly, *M. smegmatis* MC^2^ 155 was cultured at 37 °C for 36 h and diluted to 10^6^ CFU/mL in a Nutrient-Bertani broth (1% tryptone, 0.3% beef extract powder, 0.5% NaCl, and 0.05% Tween-80) at 37 °C for 36 h.

### 3.7. In Vitro Cytotoxicity

The cytotoxicity of **1** was evaluated by an MTT assay (Appendix A). In brief, the two cell lines were seeded at a density of ~5000 cells per well in 96-well plates. After 24–72 h, the cells were treated with different concentrations of the tested compounds. After further incubation for 72 h, the cell survival was determined by an addition of Cell Counting Kit-8 solution (10 μL/well). The absorbance was measured at 450 nm.

## 4. Conclusions

In this study, we established a transwell bioassay to discover LOB A (**1**), LOB B (**2**), and LOB H8 (**3**) from *S*. sp. CB09030, while a bioinformatic analysis revealed the putative LOB BGC in *S*. sp. CB09030, with a slightly different glycosyltransferase to the previously reported LobG1. LOB H8 (**3**) was directly isolated from the *S*. sp. CB09030 wild-type strain, which was previously obtained only through its heterologous expression in *S. pactum* SCSIO 02999. Our study also established, for the first time, the antibacterial activities of **4** against *M. smegmatis* and *B. subtilis*, which allow us to study the structure–activity relationships of **1**–**4**. The oligosaccharide moieties in **1** and **2** are essential for the anti-*B. subtilis* activity, while the spirotetronate aglycon is more important in the observed anti-mycobacterial activity. Since both **1** and **2** exhibited stronger anti-inflammatory activities than the clinically used indomethacin, by reducing the edema by 84–86% in the phorbol-myristate-acetate-induced mouse ear mode [8], these compounds may hold great promise against *M. ulcerans*, since they have both anti-inflammatory and anti-mycobacterial activities. The evaluation of the anti-inflammatory activities of **3** and **4** and a mechanistic study on this anti-inflammatory activity will be reported in due course.

## Figures and Tables

**Figure 1 molecules-28-03597-f001:**
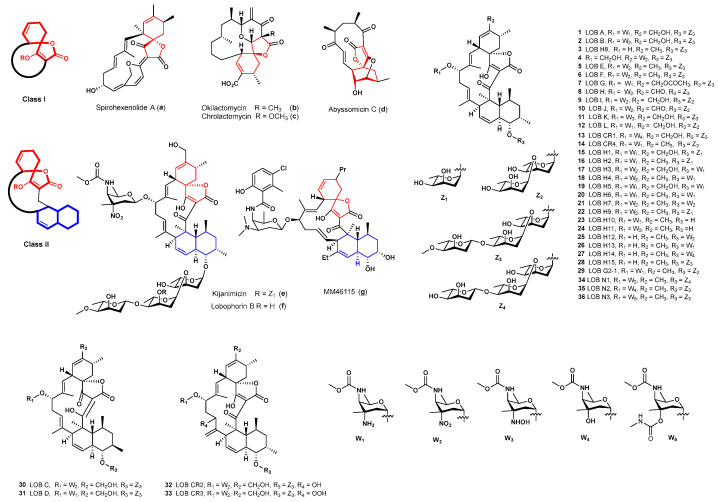
Selected representative spirotetronates and LOB derivatives.

**Figure 2 molecules-28-03597-f002:**
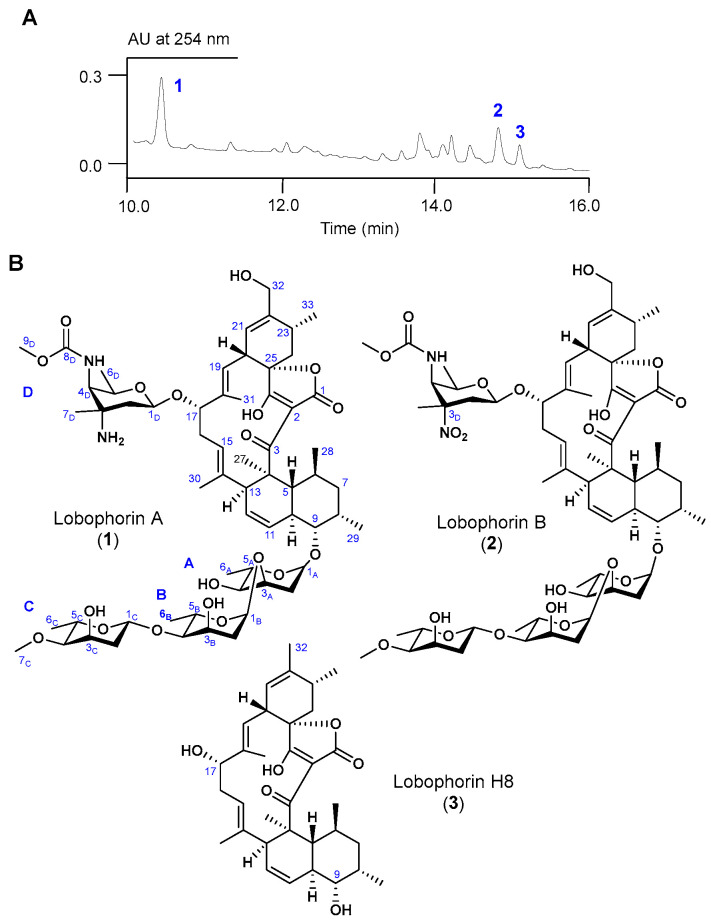
HPLC of the crude extract and chemical structure of the isolated LOBs **1**–**3**. (**A**) HPLC analysis of *S*. sp. CB09030 culture extract. (**B**) The structures of isolated **1** (LOB A), **2** (LOB B), and **3** (LOB H8) from *S*. sp. CB09030.

**Figure 3 molecules-28-03597-f003:**
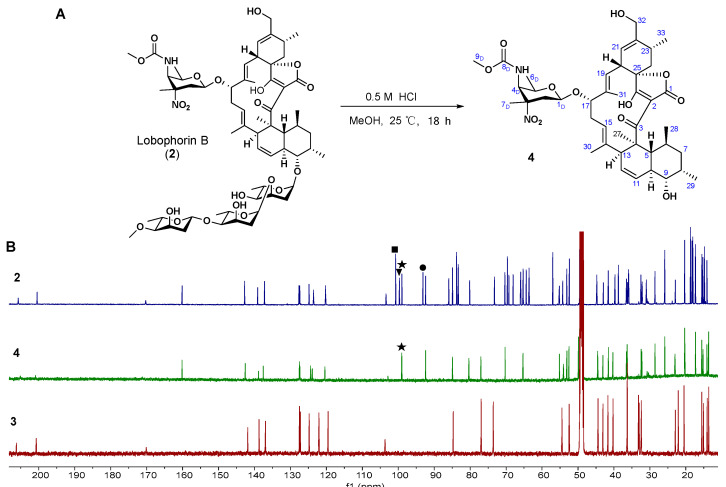
Generation of **4** from LOB B (**2**). (**A**) The hydrolysis of **2** in 0.5 M-methanolic HCl. (**B**) The comparison of ^13^C NMR spectra of **4** to **2** and **3** in CD_3_OD. ◾: The carbon signal (*δ_C_* = 100.7 ppm) from sugar C; ▼: The carbon signal (*δ_C_* = 99.7 ppm) from sugar A; ★: The carbon signal (*δ_C_* = 99.0 ppm) from D-kijanose; and ●: The carbon signal (*δ_C_* = 93.1 ppm) from sugar B.

**Figure 4 molecules-28-03597-f004:**
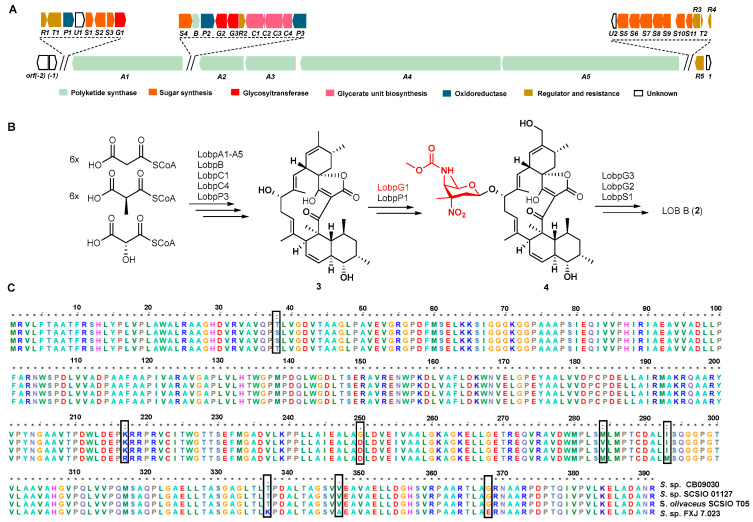
Bioinformatic analysis of LOB BGC in *S*. sp. CB09030. (**A**) Genetic organization of LOB BGC in *S.* sp. CB09030. (**B**) Proposed biosynthesis for **1**–**3**. (**C**) Sequence alignment of LobG1 from *S.* sp. CB09030, *S.* sp. SCSIO 01127, *S. olivaceus* SCSIO T05, and *S.* sp. FXJ 7.023 involved in the transferase of D-kijanose, with the varying amino acids boxed.

**Table 1 molecules-28-03597-t001:** The antimicrobial activities of **1**–**4** (MIC, µg/mL).

Compounds	*M. smegmatis* MC^2^ 155	*B. subtilis* 62305
**1**	8	8
**2**	4	1
**3**	8	32
**4**	8	32
ampicillin	-	0.5
rifampicin	2	-

-: No cytotoxicity observed.

## Data Availability

Data generated in the process of this research are available in the Appendix A.

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
