# Peer review of "The Isolation and Structure Elucidation of Spirotetronate Lobophorins A, B, and H8 from Streptomyces sp. CB09030 and Their Biosynthetic Gene Cluster"

_molecules, 2023, doi:10.3390/molecules28083597_

Round 1
Reviewer 1 Report
This current manuscript describes the isolation and structure elucidation of spirotetronate lobphorins from Streptomyces sp.
The main hypothesis of this study is to determine whether the transwell assay system can be used to screen for promising anti mycobacterial activity from some in house acetinomycete strains collected. It was reported that out of a panel of 16 in house Streptomyces strains, three lobphorins (ie, A, B and H8) were discovered. Another fourth analogue was discovered through the acid catalyzed hydrolysis of lobphorins B. Additionally the authors have also identified the biosynthetic gene cluster through genomic sequencing and bioinformatics analysis which further decipher the potential pathway involved in producing the lobphorins molecules. I would consider the topic relevant in the study of antimicrobial against Mycobacterium. There are plenty of antimicrobials identified against mycobacterium and bacillus, however, to be able to screen for these lobphorins molecules and further identified their potential biosynthetic pathway and subsequently validate their antibacterial activities in one single manuscript is considerably applaudable.Although the current published molecules are not absolutely novel in terms of their structural moieties, these are published for the first time and validated for their antimicrobial activities in vitro. As for now, the methodology seems complete and suffient. Perhaps adding in the platform for sequencing may provide extra merit to the methodology.
The conclusions are consistent with the evidence provided. All the antimicrobial activities of the lobphorins molecules were tested agains Mycobacterium smegmatis and Bacillus subtilis. The data were even compared to control antibiotics. The references were sufficiently provided.
Overall, the authors did a great job in presenting and describing their work with sufficient data to describe their hypothesis. Hence, I do not feel that further revision is required except for some occasional language formatting.
Author Response
Thanks for your comments. Indeed, a further improvement for the
methodology is to integrate genome sequencing for the tested strains, especially considering the affordable price for genome sequencing and available bioinformatic tools. We are currently pursuing this goal and would like to report it in due course.
Thanks again for both the editor and Reviewers for the rapid and critical review of our manuscript. Your comments have greatly improved the manuscript and we truly appreciate it.

Reviewer 2 Report
The manuscript presents interesting and valuable analyzes developed using modern methods. The English language should be improved and I suggest re-reading the text by a specialist native speaker. The very rich supplementary material deserves recognition. The antibacterial activity could be confirmed by some microscopic method for the selected agent.
Author Response
Thanks for your comments for our manuscript, especially the supplementary material. We then integrate the original Figure S1 into Figure 1 in the new submission, based on suggestions from you and Reviewer 3. We then carefully proof-read the whole manuscript and supplementary material.
It is an excellent idea to confirm the antibacterial activity by selected compounds with superior activities, which may facilitate the study of their mode of action. We prefer to set it aside for a promising future project.
We also carefully proof-read our manuscript and corrected some grammar errors. Some main changes are listed in the following table and also highlighted in the main text.
Thanks again for both the editor and Reviewers for the rapid and critical review of our manuscript. Your comments have greatly improved the manuscript and we truly appreciate it.
The main modification content in revised manuscript
|
Position |
Original sentence |
Modified sentence |
|
Title |
Isolation and structure elucidation of spirotetronate lobo-phorins A, B, and H8 from Streptomyces sp. CB09030 and its biosynthetic gene cluster. |
Isolation and structure elucidation of spirotetronate lobo-phorins A, B, and H8 from Streptomyces sp. CB09030 and their biosynthetic gene cluster. |
|
Line 32-35 |
spirohexenolide A [1], okilactomycin [2] and chrolactomycin [3,4]), as well as antiinflammatory and antibacterial activities (kijanimicin and MM46115 and abyssomicin C). |
spirohexenolide A (a) [1], okilactomycin (b) [2], and chrolactomycin (c) [3,4]), as well as anti-inflammatory and antibacterial activities (abyssomicin C (d), kijanimicin (e), lobophorin B (f), and MM46115 (g)). |
|
Line 42-44 |
16 LOBs have been isolated, including LOBs A-L [8-16] and LOBs CR1-CR4 (Figure S1) [17,18]. Further, 16 LOBs derivatives, including LOBs H1-H15 and LOB G2-1 (Figure S1). |
16 LOBs have been isolated, including LOBs A-L (1, 2, 30, 31, 5-12) [8-16] and LOBs CR1-CR4 (13, 14, 32, 33) (Figure 1) [17,18]. Further, 16 LOBs derivatives, including LOBs H1-H15 (3, 15-28) and LOB G2-1 (29) (Figure 1). |
|
Line 47-48 |
three derivatives LOBs N1-N3 were isolated from the mutant by disruption of P450 monooxygenase LobP1 from Streptomyces sp. SCSIO 01127. |
three derivatives LOBs N1-N3 (34-36) were isolated from the mutant by disruption of P450 monooxygenase LobP1 gene from Streptomyces sp. SCSIO 01127. |
|
Line 71-72 |
The LOB analogue 4 (O-β-d-kijanosyl-(1®17)-kijanolide) was also obtained through acid-catalyzed hydrolysis of 2 in its structure elucidation. |
The LOB analogue 4 (O-β-d-kijanosyl-(1®17)-kijanolide) was also obtained through acid-catalyzed hydrolysis of 2. |
|
Line 72-74 |
The structure-activity relationship study of 1 ̶ 4 revealed the essential role of the ol-igosaccharide moiety in the antibacterial activity of 1 and 2 against Mycobacterium smegmatis MC2155, |
The study of 1 ̶ 4 revealed the essential role of the oligosaccharide moiety in the antibacterial activity of 1 and 2 against Mycobacterium smegmatis MC2 155, |
|
Line 162-163
|
It remained to establish the importance of these amino acids for the activity of LobG1 in LOB biosynthesis, |
The importance of these amino acids for the activity of LobG1 in LOB biosynthesis remained to be established, |
|
Line 166
|
The Structure-Activity Relationship of 1 – 4 |
The Antibacterial Activity of 1 – 4 |
|
Line 167-169
|
the availability of these compounds in the current study provides an exciting opportunity to study their structure-activity relationship using simple antibacterial assays. |
the availability of these compounds in the current study provides an exciting opportunity to study their antibacterial activity. |
|
Line 206-208
|
Sixteen Streptomyces spp. from our in-house strains collection, |
Sixteen Streptomyces strains used in the current study were collected and maintained by National Engineering Research Center of Combinatorial Biosynthesis for Drug Discovery (Changsha, Hunan 410011, PR China), |
|
Line 248-251
|
Compound 2 (9.0 mg) was hydrolyzed in 0.5 M-methanolic HCl at 25 °C for 18 h to afford 4 (3.4 mg) by semipreparative RP - HPLC.
|
Compound 2 (9.0 mg) was hydrolyzed in 0.5 M-methanolic HCl at 25 °C for 18 h. The reaction system was concentrated by rotary evaporation, and then dissolved with an appropriate amount of methanol. Finally, the hydrolytic product 4 (3.4 mg) was separated by semipreparative RP - HPLC. |

Reviewer 3 Report
The authors describe the transwell-based discovery of Lobophorins (LOBs) from a panel of in-house Streptomyces strains. They have successfully isolated LOB A (1), LOB B (2), and LOB H8 (3). A novel LOB analogue 4 (O-β-D-kijanosyl-(1→17)-kijanolide) was obtained through acid-catalyzed hydrolysis of 2.
These LOBs have shown moderate antibacterial activities against Mycobacterium smegmatis and Bacillus subtilis.
Although the isolation of these LOBs is laudable, they are not novel compounds (except for structure 4). Also the characterization to confirm their structure needs more work.
I recommend major revisions and be reconsidered after all the revisions have been addressed.
Below are my comments and suggestions.
Lines 38-46- LOBs are described here. I suggest you include the figure in supporting info Figure S1 with numbering (merge with Figure 1 or have a separate figure) and when describing them in this paragraph, describe them with a compound number.
Line 58- you mentioned it would be good to have compounds with activity against Mycobacterium ulcerans. This should have been done in this study. If not, why not?
Line 92- “The 1H and 13C NMR data of 1 is almost identical to LOB A” where is this data? Include the comparison of 1H and 13C NMR of literature values of compounds 1, 2, 3 and the NMR values you obtained in a table. It should be there to confirm the structures of 1, 2, 3.
Also have you got HPLC purity levels for compounds 1~4? It should be included to confirm its purity.
Figure 2- You have labelled compound numbers for LOB A but not for 2, or 3. Label the compounds with compound numbers for 1, 2, 3 and also 4 in figure 3 so the readers can follow your NMR table S1 in SI.
2.5 The Structure-Activity Relationship of 1 – 4- This section is not really an SAR. This section is more biological testing showing the MIC values in two cell lines. You cant determine meaningful SAR trend from such a limited data set. Also what was the reason for choosing Mycobacterium smegmatis and Bacillus subtilis for antibacterial testing? This should be discussed. Have you tested on in any other cell lines? Also these compounds should be tested for mammalian cytotoxicity such as the VERO assay to determine if they could be toxic to humans if they were to be further developed into antibacterials.
4.4 Acid Hydrolysis of 2- provide more details on the synthesis of 4. How much 0.5M methanolic HCL was used, equivalence and also detailed workup procedure.
Line 270- The cytotoxicity of 1 was evaluated. Where is this data? Also why is this done for compound 1 only? It should be done for all 4 compounds.
Author Response
1. Lines 38-46- LOBs are described here. I suggest you include the figure in supporting info Figure S1 with numbering (merge with Figure 1 or have a separate figure) and when describing them in this paragraph, describe them with a compound number.
Author’s response: Thanks for your comments and we have merged LOB derivatives from Figure S1 in the supporting information with the original Figure 1 to make a new Figure 1. We also re-numbered these compounds in the revised Figure 1.
2. Line 58- you mentioned it would be good to have compounds with activity against Mycobacterium ulcerans. This should have been done in this study. If not, why not?
Author’s response: It’s an excellent point. However, Mycobacterium smegmatis MC2 155 was used as a surrogate due to biosafety concerns of Mycobacterium ulcerans in the lab. 、
3. Line 92- “The 1H and 13C NMR data of 1 is almost identical to LOB A” where is this data? Include the comparison of 1H and 13C NMR of literature values of compounds 1, 2, 3 and the NMR values you obtained in a table. It should be there to confirm the structures of 1, 2, 3.
Author’s response: Thanks for your suggestions and we included the NMR data comparison of compounds 1-4 in Table S1-S4 in the supplementary material.
4. Also have you got HPLC purity levels for compounds 1~4? It should be included to confirm its purity.
Author’s response: The purity of compounds 1~4 in HPLC or UPLC is now added in the supplementary material Figure S2.
5. Figure 2- You have labelled compound numbers for LOB A but not for 2, or 3. Label the compounds with compound numbers for 1, 2, 3 and also 4 in figure 3 so the readers can follow your NMR table S1 in SI.
Author’s response: Agreed and corrected.
6. 2.5 The Structure-Activity Relationship of 1 – 4- This section is not really an SAR. This section is more biological testing showing the MIC values in two cell lines. You cant determine meaningful SAR trend from such a limited data set. Also what was the reason for choosing Mycobacterium smegmatis and Bacillus subtilis for antibacterial testing? This should be discussed. Have you tested on in any other cell lines? Also these compounds should be tested for mammalian cytotoxicity such as the VERO assay to determine if they could be toxic to
humans if they were to be further developed into antibacterials.
Author’s response: Excellent points. Mycobacterium smegmatis and Bacillus subtilis were selected, based on previous reports for this family of compounds. There were also reports for their cytotoxicity. For example, LOB B was used to inhibit oral squamous cell carcinoma cell growth [1]. In addition, in vivo anti-inflammatory test of both LOB A and B in a Phorbol/Myristate/Acetate-induced mouse ear edema model suggested that they showed some anti-inflammatory activity [2]. Therefore, the toxicity of these compounds may not be obvious, which is a plus for their development as antibacterials.
7. 4.4 Acid Hydrolysis of 2- provide more details on the synthesis of 4. How much 0.5M methanolic HCL was used, equivalence and also detailed workup procedure.
Author’s response: 0.5 M refers to 0.5 equivalent concentration and reacts according to the concentration ratio of compound 2: HCL = 2:1. Compound 2 (9.0 mg) was hydrolyzed in 0.5 M methanolic HCl at 25 °C for 18 h. The reaction mixture was evaporated by rotary evaporation, and then dissolved in methanol. Finally, the hydrolytic product 4 (3.4 mg) was obtained by semipreparative RP-HPLC.
8. Line 270- The cytotoxicity of 1 was evaluated. Where is this data? Also why is this done for compound 1 only? It should be done for all 4 compounds.
Author’s response: The cytotoxicity of compound 1 was evaluated, and the relevant data are attached in Figure S6. Due to their low amount, no cytotoxicity assay was done for of 2-4.
Thanks again for both the editor and Reviewers for the rapid and critical review of our manuscript. Your comments have greatly improved the manuscript and we truly appreciate it.

Round 2
Reviewer 3 Report
I am satisfied there have been sufficient revisions made to the manuscript and can recommend publishing after the below minor corrections have been corrected.
5. Figure 2- You have labelled compound numbers for LOB A but not for 2, or 3. Label the compounds with compound numbers for 1, 2, 3 and also 4 in figure 3 so the readers can follow your NMR table S1 in SI.
Author’s response: Agreed and corrected.
The compound numbers are still missing for compounds 2, 3 (Figure 2.) and compound 4 (figure 3). Please label and include in the figure.
7. 4.4 Acid Hydrolysis of 2- provide more details on the synthesis of 4. How much 0.5M methanolic HCL was used, equivalence and also detailed workup procedure.
Author’s response: 0.5 M refers to 0.5 equivalent concentration and reacts according to the concentration ratio of compound 2: HCL = 2:1. Compound 2 (9.0 mg) was hydrolyzed in 0.5 M-methanolic HCl at 25 °C for 18 h. The reaction mixture was evaporated by rotary evaporation, and then dissolved in methanol. Finally, the hydrolytic product 4 (3.4 mg) was obtained by semipreparative RP-HPLC.
Please include how many mL's of 0.5M-methanolic HCl was used.
Author Response
1. The compound numbers are still missing for compounds 2, 3 (Figure 2.) and compound 4 (figure 3). Please label and include in the figure.
Author’s response: Agreed and corrected. Since compounds 1-4 have the same skeleton, we mainly marked the numbers with differences in figure 2. In addition, the numbers of compound 4 added in figure 3.
2. Please include how many mL's of 0.5M-methanolic HCl was used.
Author’s response: The volume of 0.5M-methanolic HCl is 128 µL and now added in the line
250.
